



# Brief Communication: Enabling Open Cryosphere Research with Ghub

Joseph P. Tulenko*[1,2], Sophie A. Goliber*[1], Renette Jones-Ivey[1], Justin Quinn[3], Abani Patra[4], Kristin Poinar[1,5], Sophie Nowicki[1,5], Beata M. Csatho[1], and Jason P. Briner[1]

[1]Department of Earth Sciences, State University of New York at Buffalo, Buffalo, NY, 14260, USA
   [2]Berkeley Geochronology Center, Berkeley, CA, 94709, USA
   [3]NASA Jet Propulsion Laboratory, La Canada Flintridge, CA, 91011, USA
   [4]Department of Computer Science, Tufts University, Medford, MA, 02155, USA
   [5] RENEW Institute, State University of New York at Buffalo, Buffalo, NY, 14260, USA
*These authors contributed equally to this work.

*Correspondence to*: Joseph P. Tulenko (jptulenk@buffalo.edu)

**Abstract.** We report on the infrastructure and scientific progress of the Ghub Project, a scientific gateway providing an open-access online platform for cryosphere researchers to publish and share tools and datasets. Ghub is designed to reduce bottlenecks and accelerate progress in cryosphere sciences. Open-access science lowers barriers, encourages collaboration,

and expands the network of participants in research, which is particularly important for cryosphere science given the major impact of climate change on the world's glaciers and ice sheets, whose demise contributes to sea level rise. We conclude with a brief synopsis of what the future holds for the Ghub Project.

## 1 Introduction

Sea level rise due to melting ice sheets and glaciers remains one of the largest climatical-change related threats to society, with

the most dramatic impacts likely affecting coastal communities throughout the Arctic and globally. Currently, an estimated 267 million people reside in the most at-risk areas at or below 2 meters above mean sea level (Hooijer and Vernimmen, 2021). Predicting sea level rise estimates with confidence is imperative for climate hazard mitigation and resilience planning in these communities and beyond. Reducing uncertainties and improving near-term sea level rise predictions hinges on collaboration across research domains (e.g., data/observational-producing researchers, numerical modeling researchers), which can be

facilitated with steadily improving, freely accessible, and sustainable computational infrastructures (e.g., Snow et al., 2023; Sperhac et al., 2021).

The cryosphere community has made significant strides in coordinating and predicting future ice sheet changes. For example, the Ice Sheet Model Intercomparison Project 6 (ISMIP6) has made immense progress in understanding the range of scenarios

for sea level rise due to ice sheet melt and is designed to explore the uncertainty in ice sheet projections (Nowicki et al., 2016). While these projections from dynamic ice sheet models represent considerable scientific progress, contributions to sea level





rise from ice sheets remains one of the largest sources of uncertainty in future global environmental projections (Fox-Kemper et al., 2023). The challenge lies in the dynamic nature of ice sheets and glaciers, which intersect with key components of the climate system: the ocean, atmosphere, and solid earth. The physics governing these interactions are complex and challenging

to model accurately, which leads to many different approaches that all need validation using data from physical observations. However, barriers to collecting observations can inhibit more rapid scientific traction on problems in sea level and glacier science. For example, it is difficult to observe the complex ways that ice sheets slide along their beds and characterize the conditions at the bed-ice interface since observing the bed of an ice sheet - in some places more than a few km thick – is only possible through remote sensing (radar and other geophysical survey methods; Morlighem et al., 2017), or in-situ in rare cases.

Thus, bed conditions and ice sheet sliding laws are only minimally constrained with observational data and contain large sources of uncertainty (MacGregor et al., 2022). Temporally, observational data of paleo-ice sheet margin fluctuations from the recent geologic past are invaluable for model validation (e.g., Leger et al., 2024), yet large gaps remain. Collaborations between research domains can help identify which data gaps are most necessary to fill.

We aim to enhance collaboration efficiency among traditionally disparate approaches to address these challenges through Ghub: a community-building scientific and educational computational infrastructure that encompasses models, observational datasets, processing tools, online simulation, and collaboration support, all freely available for use at theghub.org (Sperhac et al., 2021). With an expanding community-curated tool library, Ghub is a platform for diverse audiences, including – but not limited to – the numerical ice-sheet modeling community, those interested in interactive statistical tools for remote sensing,

the glacio-isostatic adjustment community and those conducting data-model comparison efforts. Here, we provide an overview of the Ghub framework and the process for users to host Jupyter Notebook-based resources such as tools, workflows, datasets and educational content on the Ghub platform. Users can host resources that require a range of computing power and storage from simple data plotting exercises to analyses that work with large datasets and require high-performance computing (HPC) resources available at the University at Buffalo and nationally. We showcase several resources already available on Ghub and

conclude by describing future goals of the Ghub Science Gateway.

## 2 The Ghub Platform

Ghub is supported by the US National Science Foundation and EarthCube and is powered by the HubZero platform developed at UC San Diego, which facilitates collaboration and the sharing of resources in a given scientific community. The resulting online environment, referred to as a science gateway, is a centralized platform for a scientific research community and beyond

– including the general public as science gateways are open-access. These gateways offer web-based interfaces tailored to specific research communities, enabling easy access to curated tools, datasets, and other resources including HPC. The objective is to provide a user-friendly interface so that even those without computational expertise can access advanced tools and high-performance computing resources. Gateways provide a depot for sharing code and computational tools on a





centralized platform, which minimizes duplication of effort and improves efficiency by streamlining development. A prime
example is nanoHUB (nanohub.org), which has been providing tools and resources to millions of users from the
nanotechnology field since 2002. While the cryosphere community is more specialized, our goal with Ghub is to deliver the
same level of service and collaboration.

The users and communities of Ghub are worldwide and range from students to senior researchers, as well as public K-12
teachers. Given our primary objective of reducing bottlenecks in scientific progress by enhancing accessibility to ice sheet
research, the Ghub team prioritizes engagement at every audience level.  Since the launch of Ghub on October 1, 2020, we
have grown to approximately 1250 users on the platform. To engage with the cryosphere community and expand the Ghub

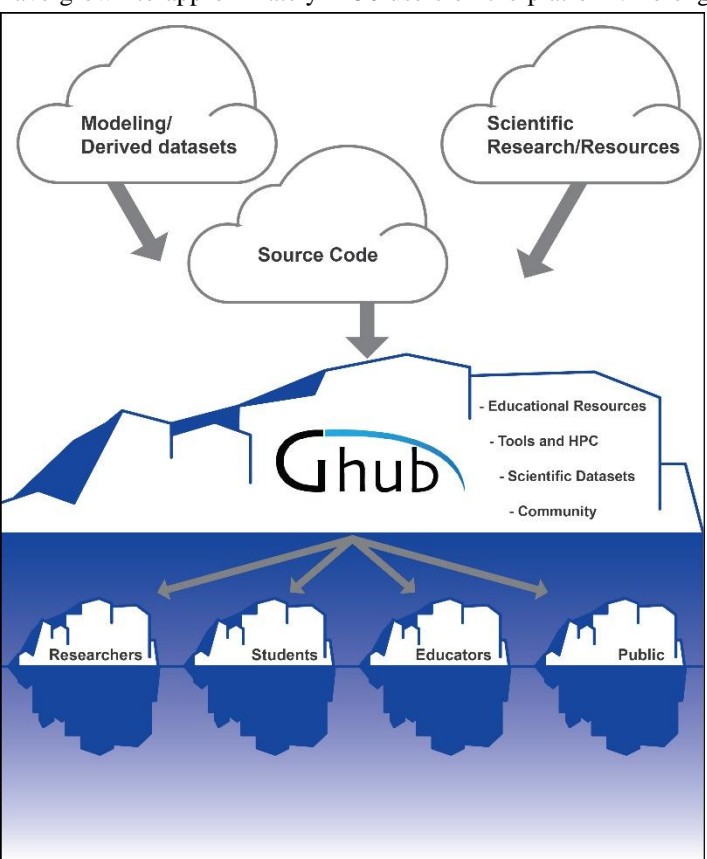

userbase, the project uses a limited team of technical support staff to help researchers publish tools. Another key component of our user base expansion strategy involves participating in conferences and hosting workshops to publicize Ghub operations and further train users. Finally, the Ghub platform can help boost visibility and engagement with both independent and Ghub platform-hosted ice sheet science educational resources by advertising those resources on the platform (see section 3.3).

**Figure 1: Conceptual model of the Ghub platform. The community provides datasets, tools and workflows, and educational content (precipitation that accumulates), Ghub provides an open-access platform for community members to publish their resources (the ice sheet), and disseminates those resources to other researchers, students, educators and the public (icebergs).**

The benefits for researchers are readily apparent, as we provide a platform to disseminate scientific work through tools, data sharing, and resources. We categorize our tools and datasets into three main domain disciplines relevant to ice sheet research: (i) paleo-observations/modelling – research on past (centennial/millennial) climate and ice sheet histories
encompassing both observational (e.g., Leger et al., 2024) and numerical modelling (e.g., Khrulev et al., 2023); (ii) projection
modelling – projects related to simulating future projections of ice sheet change, contributions to sea level rise, and model
intercomparison efforts (e.g., ISMIP6; Nowicki et al., 2016); and (iii) remote sensing/historical observations - all data and
tools related to satellite (ICESat2; Smith et al., 2023), airborne (Operation Ice Bridge; Koenig et al., 2010), and other
measurements of ice sheet changes from the recent past to today. While these categories are broad, they encompass the three



"types" of scientists who utilize Ghub. Ghub also provides a platform for educational content, including classroom-ready exercises and visualizations that focus on both general Earth-science and cryosphere-based research topics (see sections 3.3 and 4.4). Educational content accessible through the Ghub platform is developed both on the Ghub platform and externally.

## 3 Publishing Tools, Workflows, Datasets and Educational Content on Ghub

### 3.1 Tools and Workflows

Tools and workflows on Ghub span a broad range of utilities designed to benefit the polar scientific community, including data/statistical analyses, numerical simulations, and visualizations of cryosphere-related resources. All tools are built on the Jupyter Notebook interactive web application. These tools are deployed in two modes: notebook mode and application mode. Notebook mode presents the raw notebook with all code visible, which is the default mode for Jupyter notebook-based tools. Application mode involves the creation of a user interface (UI) by developers, concealing the underlying code from users.

Instead, users interact with basic UIs that allow them to upload, analyze, and download data or figures generated by tools and workflows hosted completely within the Ghub environment. Tools are commonly developed with Python coding language environments, but the Ghub platform can accommodate users coding in both R and Octave as well. The process for contributing a tool/workflow to Ghub involves a few steps with support from Ghub support staff as needed (summarized in Figure 2). The Ghub platform uses the Git distributed version control system providing a complete repository for the tool that users can

interact with both within and outside the Ghub platform. Users specify criteria for the tool – for example, where source code is stored (e.g., on GitHub or on the Ghub platform itself) and accessibility limits to the tool and source code. Users can develop code within a Ghub environment or provide pre-existing code. Once code is uploaded, users and Ghub staff can test functionality and de-bug directly on the platform. After testing is satisfactorily concluded, users can approve the tool and Ghub admin will officially publish the tool. Users can also specify how Ghub should handle licensing of the tool and request a DOI

for the tool (see 3.4).

As HubZero science gateways are open-access and computational costs scale with traffic and usage, Ghub members that generate computationally intensive tools and workflows may be required to utilize external HPC resources available through the platform. To enable contributions that require significant processing power – either through utilizing computationally

intensive workflows or analysing large sets of data – the Ghub platform offers access to high-performance computing through the University at Buffalo's Center for Computational Research (UB CCR). Tools that require access to UB CCR resources are initialized on the Ghub platform, where code is packaged and then submitted to an allocation on the UB CCR cluster. The code is run on CCR, and results are returned directly to the tool user's personal directory on Ghub. For very high resource needs, alternatives involving remote HPC use from NSF or other resources are also possible. For complex workflows with large data

transfers between stages and data-compute integration, Ghub also provides support for the Pegasus workflow engine (e.g., Jones-Ivey et al, 2025).



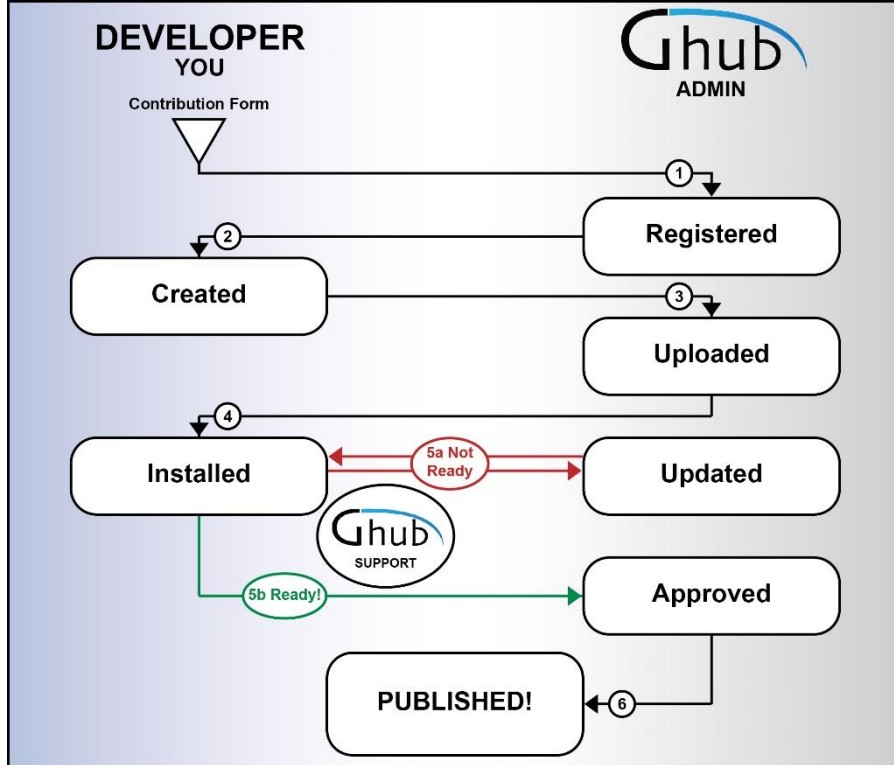

**Figure 2: Flowchart for publishing tools and workflows on Ghub. The process is initiated when individual developers fill out a contribution form provided on the Ghub platform. Then, in steps with Ghub admin, users 1) register their tool on the platform, 2) create a new tool based on customizable registration specifications, 3) upload code developed either offline or through a development container freely available to each registered user, work with the Ghub support staff at UB to test tools prior to publishing and commit updated code, and then 6) once approved, the tool is published.**

## 3.2 Derived Datasets

Derived datasets are datasets that are created from existing data sources. The criteria for a derived dataset typically involves a substantial alteration to the data structure while preserving the original data values (e.g., combining multiple datasets, optimizing raw data, or subsetting for a particular region). Each derived dataset is citable, but users are asked to cite the original data as well. A derived dataset example is "IceBridge ATM L2 Icessn Elevation, Slope, and Roughness" dataset (Narkevic et al., 2021), which is a resampled and smoothed elevation, slope and roughness dataset of the Greenland Ice Sheet ice surface derived from NASA's Airborne Topographic Mapper (ATM). The dataset is provided in formats common to scientific research. Several benefits arise from hosting derived datasets on the Ghub platform. Primarily, complete datasets hosted by external sources (e.g., the ATM datasets provided by NASA) can either be i) large (up to terabytes in size) making them difficult to navigate and then apply to specific tasks, and/or ii) datasets can come in formats that are difficult to manipulate. In either case, it can be beneficial to utilize the Ghub platform to host derived datasets where subsetting, combining and post-processing methods are performed on externally hosted datasets. As with tools and workflows that require high-performance computing resources, large datasets that exceed the HubZero platform storage limitations can be accommodated through UB CCR. These datasets are intended typically for broad community-wide use (see for example the ISMIP6 datasets described in section 4.1). The datasets are served through Globus endpoints, which are commonly used across numerous scientific domains for serving large datasets.

## 3.3 Educational Content

The Ghub platform is intended to be a 'one-stop-shop' for those interested in ice sheet science. To appeal to the broader audience of science educators, we provide two general options to boost visibility and engagement with cryosphere-related



educational resources. Educators can either (i) develop and host their own educational content on the Ghub platform, or (ii) use Ghub to host links to external resources maintained outside the Ghub platform. Resources developed on Ghub can range from teaching lab modules that utilize computational workflows (see section 4.3) to interactive visualizations of aspects of ice sheet research (e.g., Goliber and Christian, 2024). Like tools and datasets hosted on the Ghub platform, the Ghub team can provide any educational resource with licencing and DOIs linked to the resource (see section 3.4). Users interested in utilizing

Ghub to boost engagement with externally hosted resources can provide links to the Ghub platform from virtually any online-hosted resource related to ice sheet science.

### 3.4 Code and data availability, licensing, and DOIs

The Ghub platform provides users with several options for making their resources available and citable. Source code for tools and educational resources developed on Ghub can be specified as open or closed, and contributors can specify the licencing

for any resource (tools, datasets and/or educational content) developed through the platform. The Ghub team currently recommends either the General Public License v3.0 (GNU GPLv3), which only prohibits platform visitors from distributing closed-source material, or the MIT Licence, which allows visitors to distribute both open- and closed-source material. For information about licencing, the Ghub team directs readers to the 'choose an open-source licence' resource available at https://choosealicense.com/. Ghub team members also encourage users to generate citable DOIs for resources hosted on the

Ghub platform. Users can link pre-existing DOIs on their own for resources and specify that visitors should use the provided DOI when citing the resource. If users do not already have DOIs for their contributed resources, the Ghub team can generate Zenodo DOIs for any resource hosted on the platform. Ghub-provided DOIs can likewise be associated with the resource and visitors can use the Zenodo DOI when citing the resource.

### 4 Highlighted resources currently available at Ghub

#### 4.1 ISMIP6 Projections for Greenland and Antarctica

The Ice Sheet Model Intercomparison Project for CMIP6 (ISMIP6) is a large collaborative effort between ice sheet and climate modelers along with experts in polar observations, with the goal to assess contributions of melting ice sheets to future sea level rise using an ensemble of community ice sheet simulations (Nowicki et al., 2016). The project was a subset of the larger Coupled Model Intercomparison Project -phase 6 (CMIP6) and was the first time in CMIP history that an effort was dedicated

to projections of ice sheet evolution. Although the impacts of numerous physical processes and mechanisms driving ice sheet behavior remain uncertain, especially as scientists project ice sheet behavior past 2100 CE, recent efforts to assess the spread in uncertainty in an ensemble of simulations help highlight where the largest sources of uncertainty remain (Seroussi et al., 2024). The numerical results from the large ensemble of ice sheet simulations that participated in ISMIP6 Antarctica 2300 CE projections require storage on the magnitude of terabytes – a considerable challenge for handling and disseminating data for

various analyses. The ISMIP6 team contributed Globus Endpoint resources to Ghub (hosted on UB CCR) allowing public




access to the ensemble data and forcings for all experiments as well as tools to analyse these data to solve this challenge. For example, Antarctica 2300 simulations can be cited as Nowicki and ISMIP6 team (2024). Access to the complete dataset via Globus and tools on Ghub allow researchers interested in analysing ISMIP6 results to run more efficient analyses and ultimately speed up informing the cryosphere community where research efforts need to be focused going forward.

## 4.2 The ATM-Based Crevasse Detection and Extraction workflow

NASA's Airborne Topographic Mapper (ATM) instrument onboard the Operation Ice Bridge (OIB) mission collected high-resolution laser altimetry data over Earth's glaciers and ice sheets. Designed to sense the surface height over wide areas, the ATM data is also suitable for observing small structures in the ice surface, including crevasses. Ghub contributors developed the ATM-Based Crevasse Detection and Extraction (ABCDE) tool, which detects local negative height anomalies in the ATM data associated with ice surface crevasses. With appropriate user-entered parameters like maximum width and minimum depth, this tool will locate and isolate likely crevasses. The tool is currently available and citable on Ghub (The ABCDE Tool; Jones-Ivey et al., 2025). The tool was used in a recent study to highlight the ubiquitous concurrence of crevasses at the downstream boundary of firn aquifers along the southeast section of the Greenland Ice Sheet. Firn aquifers – reservoirs of meltwater stored and insulated just below the surface of ice sheets commonly observed today – near the margins of the Greenland Ice Sheet can provide significant quantities of meltwater to the base of the ice sheet, which has important implications for the dynamic process of ice sheet basal sliding (Cicero et al., 2023).

## 4.3 The Introduction to Python for Earth Science Educational Resource

Several computer programming languages enable Earth scientists (including those in the cryosphere community) to perform shared and open data analyses, visualizations, and numerical simulations in efficient ways to contribute to our understanding of Earth system processes. Some of the most common languages are MATLAB, Python and R, which all have strengths and weaknesses depending on how they are applied. Ghub contributors recently developed an introductory course designed to cover the basics of utilizing the Python programming language for applications in Earth Science. Lesson topics include packages designed to manipulate geospatial data (e.g., geopandas and xarray), and conduct statistical analyses (e.g., scipy), among other packages. Python has several self-evident benefits for Ghub members, especially given that libraries maintained by the global community of Python contributors are open-sourced – following principles shared by the Ghub platform – and that Ghub provides Jupyter Notebook-based Python environment containers for users to develop, test and publish code. The resource is intended to reach a broad audience of students, professional geologists, researchers, and enthusiasts alike interested in quickly learning programming skills directly applicable to Earth science. The educational resource is currently available and citable on the Ghub platform (Intro2Python; Goliber et al., 2025).





## 4.4 The AskICE-D Tool

Cosmogenic-nuclide exposure dating is a critical geologic dating method to constrain earth surface processes, which includes past ice sheet fluctuations that are of particular interest for the cryosphere community. These observations highlight climatic forcings and processes that drive ice sheet and glacier change and provide benchmarks for paleo-ice sheet simulations to assess ice sheet model accuracy. The Informal Cosmogenic-Nuclide Exposure-Age Database (ICE-D) Project computational infrastructure (Balco, 2020) is a recent, community-wide effort to efficiently store and dynamically serve cosmogenic-nuclide datasets. The Google Cloud-hosted infrastructure stores a suite of sample-specific observations used to calculate meaningful geologic parameters (e.g., ages of past ice margin positions) in relational database. Ages can be dynamically served for synoptic-scale workflows, visualizations and analyses. The dynamic nature of the computational infrastructure accommodates both (i) new datasets, and (ii) improvements to complex age calculations. Access to the ICE-D computational infrastructure is open to anyone interested in utilizing the data yet requires some programming experience. To facilitate engagement with the ICE-D project and lower barriers to entry for interested users, Ghub contributors developed a tool with a simple user-interface with options for filtering parameters of cosmogenic-nuclide datasets stored in ICE-D (The AskICE-D tool; Tulenko et al., 2024). These user-defined parameters then build queries that are dynamically sent to ICE-D, which in turn serves the requested data in an easily manipulated table. Users can take these datasets offline for personal use, but concepts of dynamically filtering ICE-D data demonstrated in the tool can also be applied to other computational workflows hosted both on Ghub and elsewhere.

## 5 The Future of Ghub

The Ghub platform is steadily growing via an increasing number of contributors and members spanning many domains in cryosphere sciences. The vision for continued expansion of the Ghub platform lies primarily in three areas: 1) continue to engage with motivated cryosphere community members to contribute resources to the Ghub platform and expand the userbase – this can be addressed by hosting additional workshops and training events, presenting at professional conferences and meetings, and through personal communication; 2) maintain efforts to engage specifically with non-experts in cryosphere research by promoting educational content on the Ghub platform – this is done through both creating more resources (such as the Introduction to Python resource) and providing a platform for users to discover other educational resources hosted externally from Ghub; 3) encourage and facilitate the transition from hosting disparate, expensive, post-processed datasets to hosting resources that 'bring the code to the data'. A goal of the Ghub team is to accommodate already-existing datasets hosted on UB CCR (e.g., ISMIP6 2300 AIS projections) and elsewhere (e.g., NASA EarthData and ICE-D). Similar platforms have adopted this approach (e.g., CryoCloud; Snow et al., 2023) and the practice leads to more efficient workflows and a reduction in the dependence on considerable data storage costs. A successful Ghub will also act as a technology scout for the cryoscience community engaging with emerging methodologies (e.g. AI based modeling) and commission tools from members and Ghub investigators for timely and low barrier adoption by the community.

## 6 Conclusions

The Ghub Science gateway is currently facilitating critical cryosphere research through an open-source online platform. The platform lowers barriers for researchers to both access and host tools and datasets and importantly provides a space for new and rising scientists to discover scientific and educational resources as well. Efforts to efficiently serve cryosphere community-wide resources are crucial for expediting collaborative efforts to assess the impact of changing climate on ice sheets and glaciers world-wide.

## Data and Code availability

All code, data, and computing environments discussed in this brief communication are available via Zenodo citations and on the Ghub platform (theghub.org).

## Author contributions

JPT: conceptualization, writing (original draft), visualization, project administration. SAG: conceptualization, writing (original draft), project administration. RJ-I: conceptualization, writing (review and editing), project administration. JQ: conceptualization, writing (review and editing), project administration. AP: conceptualization, writing (review and editing), funding acquisition. KP: conceptualization, writing (review and editing), funding acquisition. SN: conceptualization, writing (review and editing), funding acquisition, project administration. BMC: conceptualization, writing (review and editing), funding acquisition. JPB: conceptualization, writing (review and editing), funding acquisition, project administration.

## Competing interests

One of the coauthors is a member of the editorial board of *The Cryosphere*.

## Acknowledgements

We would like to thank HubZero for providing the platform that Ghub runs on. The Ghub project is supported by NSF Grant number 2004826. We would like to acknowledge the land on which the University at Buffalo operates, which is the territory of the Seneca Nation, a member of the Haudenosaunee/Six Nations Confederacy. Today, this region is still the home to the Haudenosaunee people, and we are grateful for the opportunity to live, work, and share ideas in this territory.

## Financial support

This brief communication has been supported by NSF Grant number 2004826.



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
