# Peer review of "Brief Communication: Enabling Open Cryosphere Research with Ghub"

_EGUsphere, 2025_

## Author Response (AR1)

**Response to RC1 'comment on egusphere-2025-894'**

*We thank Dr. Golledge for their supportive comment and recommendation to accept the manuscript.*

**Response to RC2 'comment on egusphere-2025-894'**

*We thank Dr. Buzzard for their insightful and constructive review. Please see our responses to each comment raised below (original review comments are in bold and our responses are in italics).*

**Line 56- Introduction of the name Ghub- I think it would be helpful here to explain the name Ghub (e.g. is the g 'glacier'?) as it is quite close to Github and has some overlap in purpose it could be confused with a Github resource.**

*Thanks for the suggestion, we agree that it would be helpful to clarify and explain the name Ghub since it is close to GitHub. We add in the following phrase:*

*Line 56: "Ghub, which stands for 'glaciology' hub not to be confused with GitHub, is supported by the...."*
* * *
**Figure 1- Should educational resources also be in cloud as they need to be contributed?**

*Fair point, the platform does allow for two pathways in terms of educational resources; (i) users can develop and host educational content directly on the Ghub platform (as is the case for the Intro2Python resource), and (ii) users can provide links to externally-hosted educational resources that can be more or less advertised on the Ghub platform as is the case suggested here. We will update the figure accordingly.*
* * *
**Line 165- Should Globus endpoints have a citation or link?**

*Thank you for the opportunity to clarify this in the manuscript. That is correct, each globus endpoint has a link and each curated dataset hosted on Ghub will provide the link to its respective globus endpoint. We add in the following phrase:*

*Line 166: "... are commonly used across numerous scientific domains for serving large datasets. Links to globus endpoints are accessible on each dataset's landing page."*
* * *
**Line 202- It might be helpful to show this citation in a similar way to line 211 as a demonstration that it's not just a reference to a paper.**

*Agreed that it would be better to demonstrate that this citation is to a Ghub dataset and not a peer-reviewed paper. We re-write the sentence to:*

*Line 201: "For example, users can find comprehensive model outputs from ensemble Antarctica 2300 simulations openly accessible on Ghub (ISMIP 6 23rd Century Projections; Nowicki and ISMIP6 team, 2024)."*
* * *
**Line 225- I think these principles could be made clearer earlier on in the manuscript. The authors have done a very good job of motivating the problem, but their motivation in producing and maintaining Ghub in the way they have chosen to was less clear. Explicitly stating principles e.g. open source, FAIR etc. (or whatever they have been chosen to be) would help the community understand what this resource is, and help differentiate the purpose of this manuscript from Sperhac et al.**

*We thank our reviewer for this comment and agree that the wording in this section about specifically developing modules for Python coding because it is an open-access coding language (and aligns with the platform's guiding principles) could be highlighted sooner and thus get reiterated here. We revise the following sentence to clarify those points earlier in the manuscript:*

*Line 59: "The resulting online environment, referred to as a science gateway, is a centralized platform for scientific research communities and beyond (including the general public). Science gateways developed through HubZero are open access and tools may be made open source. Utilizing the HubZero platform achieves a core principle for the Ghub project by providing users free access to cryosphere-related tools and resources developed by and for the community."*

*we also take the opportunity to briefly reiterate that resources hosted on Ghub are openly available by revising:*

*Line 178: "The Ghub platform provides users with several options for making their resources openly available and citable."*

*Readers should also note that the manuscript (briefly) specifies that the platform is open access in a handful of lines elsewhere throughout (e.g., line 14 and line 122).*
* * *
**Line 257- It would be good to briefly mention differences between Ghub and CryoCloud here- given the purpose of the manuscript seems to be specifically to notify the cryosphere community about this resource it would be helpful to know how it related to other approaches.**

*We agree this is a good opportunity to help readers understand differences between CryoCloud and Ghub since they serve overlapping communities but have somewhat different goals. We have chosen here and elsewhere to clarify those differences:*

*Line 258: "... dependence on considerable data storage costs. Specifically, the CryoCloud platform has demonstrated success in developing environments that 'bring the code to the data' for workflows that utilize ICESat2 data, and Ghub aims to adopt this approach for a diverse range of externally hosted data portals (see for example the interoperability between Ghub and the ICE-D project)."*

*Line 112: "...users coding in both R and Octave as well. It should be noted that a primary directive of the Ghub platform is to host tools that are already operational. While there is code development space available to users on the Ghub platform, cryosphere community members are also encouraged to visit complementary platforms such as CryoCloud (Snow et al., 2023) if seeking additional support to develop their own computational tools."*